# Marine heatwaves shift ocean net primary productivity from the tropics toward the poles

Ce Bian [1] ✉, Zijie Zhao[2,3,4], Neil J. Holbrook [3,4,5], Peter G. Strutton [3,4] & Lixin Wu [6,7]

Marine heatwaves (MHWs), prolonged extreme thermal events, are reshaping ocean ecosystems, yet their influence on global productivity patterns remains poorly understood. Here, we use a global regression framework to disentangle linear thermal effect from nonlinear feedback and demonstrate that MHWs restructure the dominant drivers of ocean net primary production (NPP). MHWs induce a regime shift from sea surface temperature (SST)-independent to SST-dependent controls on NPP anomaly, reflecting an enhanced thermal effect in response to extreme warming. MHW suppressed the NPP anomaly across nutrient-poor low latitudes but increased it in nutrient-rich higher latitudes. The contrasting responses arise from differences in nutrient baselines, with low-nutrient regions exhibiting greater sensitivity to extreme warming. Together, these results reveal an emerging poleward redistribution of ocean productivity and highlight the need to incorporate MHWs into projections of marine ecosystem resilience and climate–biosphere feedbacks.

Human activities have disrupted Earth's energy balance, with nearly 90% of the excess heat stored in the upper ocean, where warming manifests across multiple spatial and temporal scales[1]. Among these manifestations, marine heatwaves (MHWs)—prolonged periods of anomalously high ocean temperature—have become a major focus of contemporary climate research[2–4]. Their prominence stems from wide-ranging ecological and socioeconomic impacts, including shifts in species distribution, increased mortality, reductions in fishery and aquaculture yields, and altered ecosystem functioning[5–7]. MHWs have increased in frequency, duration, and intensity over recent decades[8], and are projected to intensify under continued warming[9]. As climate change accelerates, clarifying how MHWs perturb marine ecosystems has become increasingly urgent.

Early understanding of MHW impacts was shaped by widespread coral bleaching events[10,11], which highlighted the vulnerability of temperature-sensitive organisms. However, ecosystem responses to warming differ substantially due to variations in physiological traits and trophic structure[12], and robust ecosystem-scale assessments remain limited by sparse observations. This motivates the need for scalable indicators of ecosystem functioning. Ocean net primary production (NPP)—the balance between phytoplankton photosynthesis and respiration[13]—has emerged as a foundational metric because it underpins marine food webs and modulates global carbon cycling[14,15]. Quantifying how MHWs influence NPP is therefore essential for evaluating ecosystem resilience and anticipating productivity shifts in a warming ocean[16].

While temperature, light, and nutrient availability are well-established drivers of oceanic NPP[17], most existing studies have focused on long-term climatological trends or species-level physiological responses[12,18,19]. In contrast, the impacts of short-lived yet intense disturbances, such as MHWs, and their potential ecological feedbacks remain poorly understood[20–23]. Accordingly, a quantitative assessment

[1]Lamont Doherty Earth Observatory, Columbia University, New York, NY, USA. [2]Department of Earth System Science, University of California, Irvine, CA, USA. [3]Institute for Marine and Antarctic Studies, University of Tasmania, Hobart, TAS, Australia. [4]Australian Research Council Centre of Excellence for Climate Extremes, University of Tasmania, Hobart, TAS, Australia. [5]Australian Research Council Centre of Excellence for the Weather of the 21st Century, University of Tasmania, Hobart, TAS, Australia. [6]Frontiers Science Center for Deep Ocean Multispheres and Earth System and Key Laboratory of Physical Oceanography, Ocean University of China, Qingdao, China. [7]Laoshan Laboratory, Qingdao, China. ✉e-mail: cebian@ldeo.columbia.edu

of how MHWs influence NPP variability is essential for understanding their broader ecological impacts. This knowledge gap is particularly critical in large marine ecosystems (LMEs)[24], which, while covering only ~22% of the ocean surface, contribute over 95% of global fisheries catch[25]. Their high ecological productivity and socio-economic importance mean that the human societies relying on LMEs are potentially vulnerable to MHW-driven perturbations, especially where human pressures further amplify ecological impacts. Addressing how NPP responds to MHWs in these regions is thus essential for anticipating future ecosystem shifts and informing sustainable management strategies.

Here we integrate satellite observations and reanalysis products to quantify the global influence of MHWs on oceanic NPP, with additional focus on LMEs. Using a regression-based framework, we separate NPP anomalies (NPPA) into SST-dependent and SST-independent components, allowing us to identify a robust regime shift in the dominant processes governing NPPA: from SST-independent variability under moderate warming to SST-dependent control during MHWs. This transition is associated with a pronounced latitudinal reorganization of productivity, characterized by suppressed NPP in low-latitude systems and enhanced NPP at higher latitudes. Together, these results reveal a latitude-structured coupling between thermal anomalies, nutrient supply, and baseline ecosystem properties, providing a mechanistic basis for anticipating ecosystem vulnerability under future ocean warming.

## Results

### Response of global oceanic NPP to MHWs

Oceanic NPP exhibits strong geographic variability, largely driven by contrasts between coastal and open-ocean regions (Fig. 1a). Higher NPP is observed along continental shelves and nearshore waters, such as the Yellow Sea, the Maritime Continent, and the Bay of Bengal, due to river input of nutrients. Elevated productivity is also evident in regions of strong upwelling, including eastern boundary currents, equatorial divergence zones, and the Arabian Sea, where nutrient-rich deep waters are brought to the surface. In contrast, subtropical gyres feature weak vertical mixing, limited nutrient supply, and typically low NPP. High-NPP regions closely align with LMEs (black outlines in Fig. 1a), which support abundant fisheries and are vital to global food security. While seasonal variability in NPP is evident[26], particularly in mid- to high-latitude regions, the spatial distribution remains broadly consistent throughout the year (Fig. S1).

MHWs induce pronounced spatial shifts in NPP, forming a distinct latitudinal gradient in NPPA: decreases in low-latitude regions and increases in high-latitude regions (Figs. 1c and S2a). This pattern resembles that observed during generally warm periods ($SSTA^+$, i.e., positive SST anomaly (SSTA) except MHWs, see "Definition of MHWs and $SSTA^+$" in Methods), but with greater magnitude under MHW conditions (Fig. 1b, c). Specifically, MHWs result in 4%–21% extra NPP increase in high latitudes (45-80°S, 45-80°N) and 4%–10% NPP reduction in low latitudes (45°S −45°N) (Fig. S2a). Positive NPPA during MHWs is especially prominent in regions influenced by strong

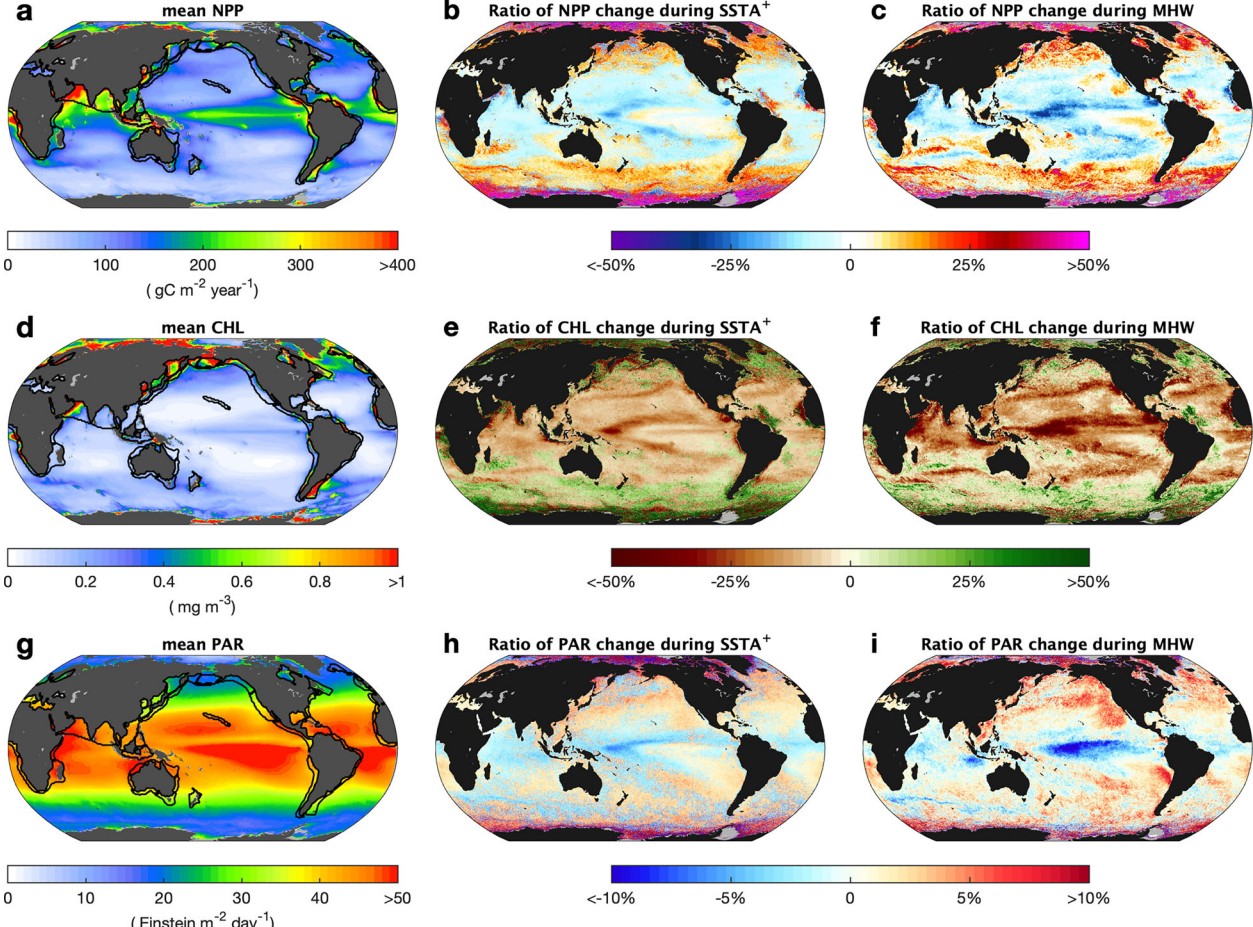

**Fig. 1 | Spatial pattern of oceanic net primary productivity (NPP) and environmental drivers during generally warm periods ($SSTA^+$) and MHWs. a** spatial pattern of mean NPP during 1988–2018, with LMEs outlined by black lines. **b, c** Ratio of NPP changes compared with their local mean value during $SSTA^+$ periods and MHWs, respectively. Same as **a–c**, **d–f** are the mean chlorophyll concentration (CHL) and its corresponding change ratio during the $SSTA^+$ and MHWs period. **g–i** represent mean photosynthetically active radiation (PAR) and its anomaly ratio during $SSTA^+$ and MHW periods.

upwelling and mesoscale dynamics, including western boundary currents and their extensions (WBCEs) and eastern boundary upwelling systems (EBUSs). The NPPA during MHWs is also prominent in the Southern Ocean, the North Atlantic, and especially the polar regions (Fig. 1c). LMEs show particularly strong responses that mean NPPA in LMEs reaching 257 gCm$^{-2}$ year$^{-1}$ during MHWs, nearly twice the global mean NPPA (147 gCm$^{-2}$ year$^{-1}$) (Fig. 1a–c). Importantly, these pattern emerges consistently across independent NPP products, including the CAFÉ model[27] (Fig. S3g–i) and the carbon-based production model (CbPM)[28,29] (Fig. S3d–f). The agreement among multiple datasets highlights the robustness of the statistical decomposition.

The underlying mechanisms driving these patterns include physical and biological components. NPP algorithms commonly incorporate chlorophyll concentration (CHL) and photosynthetically active radiation (PAR)[30] as key predictors[27,28]. Elevated SST during MHWs affects photosynthesis and respiration rates[31], altering the metabolic balance of phytoplankton. Changes in PAR indicate variations in light availability and thereby contribute to NPPA. The spatial distribution of CHL closely mirrors that of NPP, with higher concentrations in high latitudes, but lower concentrations in low latitudes (Fig. 1d). Although the reduction of CHL in low latitudes is larger during MHWs than during $SSTA^+$, the increase of CHL in high latitudes during MHWs is smaller than during $SSTA^+$ (Fig. S2b). PAR changes are similar during $SSTA^+$ and MHWs, but MHW-induced changes are more intense around 40–75°N (Fig. S2c), mainly in the northeast Pacific, a known hotspot for recurrent and intense MHWs[32,33] (Fig. 1i). These PARAs may reflect shifts in potential light limitation, although their influence is captured statistically rather than interpreted as a mechanistic causal effect. Together, these results reveal a clear latitude-dependent pattern in the response of NPP and its relationship to MHWs.

## Causal links between temperature and NPP change

Although rising SST is known to influence marine productivity[34], the extent to which observed changes reflect direct temperature-associated variability versus broader ecological adjustments remains uncertain. To separate these influences, we applied a statistical decomposition in which anomalies in CHL and PAR were partitioned into components linearly associated with SST anomalies and residual components not explained by SST (see "Statistical modeling of NPPA" in Methods). The resulting NPPA variability was grouped into SST-dependent terms—representing the linear SSTA–NPPA covariation expressed through multiple pathways—and SST-independent terms, which capture remaining variability including nonlinear ecological responses, unmodeled processes, and time-lagged effects.

Across most of the global ocean, SST-independent terms account for a greater fraction of NPPA variability during general warm ($SSTA^+$) conditions (Fig. 2b, c). These contributions are especially pronounced in coastal and upwelling systems, where NPPA responds strongly to processes that do not covary linearly with SST. In contrast, SST-dependent contributions exhibit a more heterogeneous spatial structure, with positive associations at high latitudes and negative associations in low-latitude regions (Figs. 2b and S4a). The SST-dependent component is mainly influenced by the linear SSTA term and by PARA (Fig. S5a–c), whereas SST-independent contributions are dominated by $CHLA_{res}$ and $NPPA_{res}$ terms (Fig. S5d–f).

When conditions transition into MHWs, the balance between these two components changes markedly. SST-dependent terms account for a substantially larger fraction of NPPA variability, while SST-independent contributions decrease, particularly NPPA$_{res}$ (Figs. 2d–f and S5g, l). This shift indicates a strengthened first-order covariance between SSTA and NPPA during extreme warming conditions, rather than a mechanistic dominance of temperature.

The spatial structure of this transition is further illustrated by the distribution of dominant terms (Fig. 3). Under $SSTA^+$ conditions, SST-independent processes dominate NPPA variability in 84% of the global

ocean and 94% of LMEs (Fig. 3a, b and Table S2). During MHWs, these proportions reverse, with SST-dependent terms dominating 79% of the global ocean and 77% of LMEs (Fig. 3d, e). The magnitude of this transition varies geographically. WBCEs, EBUSs, and ice-covered polar regions show strong shifts toward SST-dependent influence, whereas subtropical gyres exhibit weaker changes (Fig. 2j and Table S2). EBUSs display the largest NPPA adjustments and the broadest response ranges during MHWs, reflecting the diverse and nonlinear ecological dynamics characteristic of these regions (Fig. 2j). Because many EBUSs overlap with LMEs, their internal ecological heterogeneity further broadens their response spectra (Fig. 2k).

Taken together, these results reveal a coherent, spatially structured intensification of SSTA-associated NPPA variability during MHWs. This shift is expressed as a strengthened latitudinal gradient—suppressed productivity in the tropics and enhanced productivity at higher latitudes and across most LMEs (Fig. 2h)—representing a statistical reorganization of productivity–temperature covariance under extreme warming conditions.

## Enhanced sensitivity of LMEs to MHWs

Given their ecological and socioeconomic importance, we further examined the response of LMEs (Fig. 2k). During $SSTA^+$ periods, NPPA in LMEs ranges from −30 to 37 gCm$^{-2}$ year$^{-1}$ and is primarily driven by SST-independent processes. In contrast, MHWs yield a broader range of anomalies (−30 to 50 gCm$^{-2}$ year$^{-1}$), with a net positive skew and a greater role for SST-dependent drivers (Fig. 2k). The interquartile range during MHWs suggests an overall increase in productivity in LMEs.

However, this response is spatially heterogeneous. During MHWs, most LMEs around South America and Europe exhibit strong positive NPPA, with the upper quartile values reaching up to 48 and 60 gCm$^{-2}$ year$^{-1}$, respectively (Fig. 2k). In contrast, African LMEs experience net declines, with the first quarter value as low as −25 gCm$^{-2}$ year$^{-1}$. LMEs around North America and Asia have NPPA responses comparable to the global mean, whereas those around Australia display the least response to MHWs and no discernible difference from $SSTA^+$ conditions (Fig. 2k).

Despite these regional differences, a common pattern emerges: the dominant driver of NPPA shifts from SST-independent processes under $SSTA^+$ periods to SST-dependent processes under MHWs (Fig. 2k). Specifically, 54 of the 57 LMEs are dominated by SST-independent drivers under $SSTA^+$. Only three regions: Baltic Sea and Faroe Plateau (Europe), and the Somali Coastal Current (Africa), exhibit SST-dependent dominance (Fig. 3b). During MHWs, there are 50 LMEs dominated by SST-dependent drivers while only seven retain SST-independent dominance, including the Gulf of Mexico and Aleutian Islands (North America), East Brazil Shelf (South America), Black Sea (Europe), Guinea Current (Africa), and the Gulf of Thailand and West Bering Sea (Asia) (Fig. 3d). Notably, the 7 LMEs that retain SST-independent dominance during MHWs are those where nonlinear processes continue to exert a primary influence on NPPA, indicating that the response of these ecosystems remains strongly shaped by non-thermal dynamics.

## Drivers of global ocean NPPA responses to MHWs

To characterize spatial differences in productivity responses to extreme warming, we classified the global ocean based on the sign of the SSTA–NPPA relationship, retaining only correlations significant at the 90% confidence level. Regions with significant positive correlations are referred to as MHW-enhanced, whereas regions with significant negative correlations are categorized as MHW-inhibited (Fig. 4a). This diagnostic reveals a clear large-scale structure: high-latitude basins, western boundary current extensions, and marginal ice zones tend to fall into the MHW-enhanced category, while low-latitude subtropical and tropical regions—including nutrient-limited gyres—are primarily

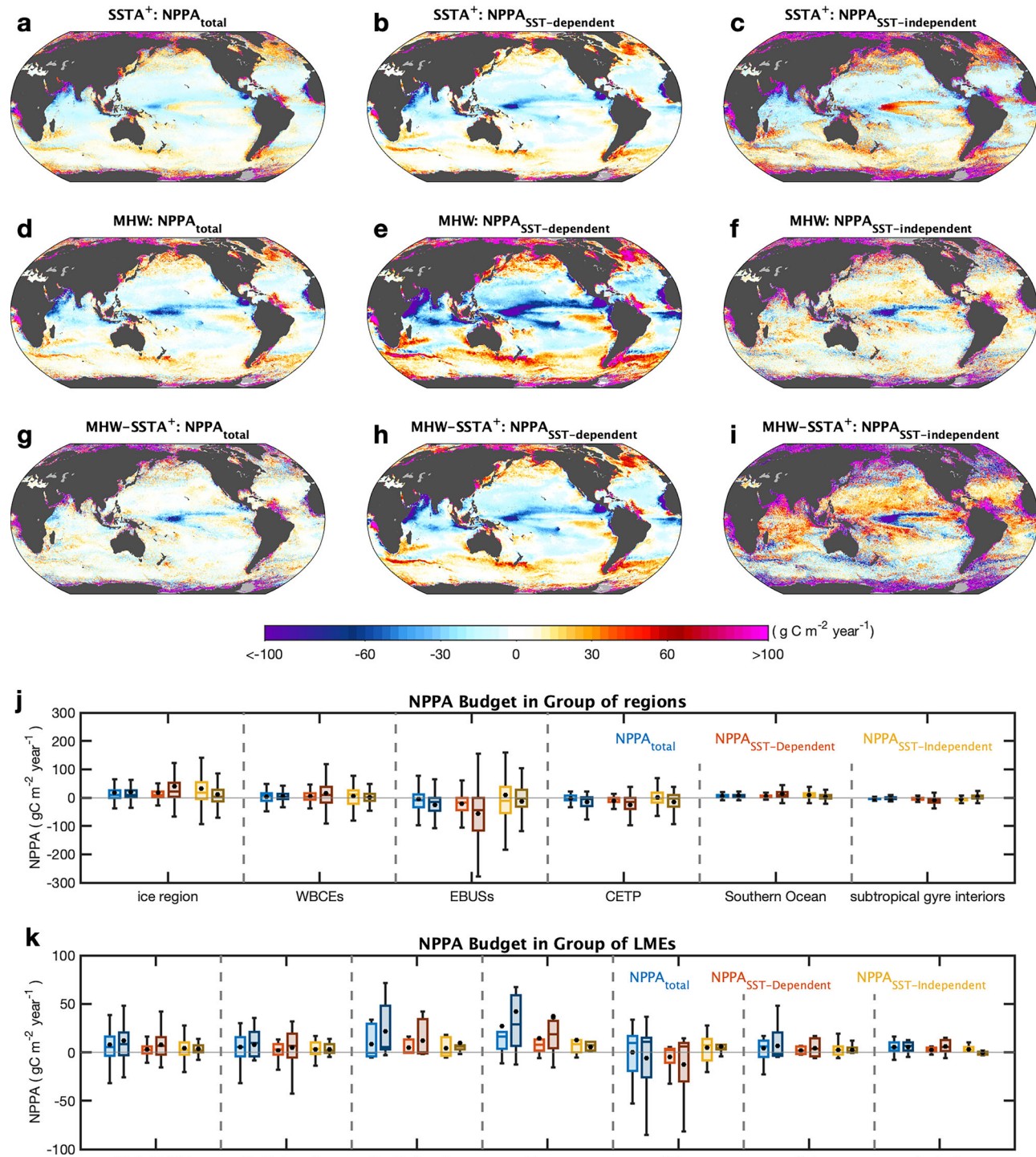

**Fig. 2 | Budget of oceanic NPPA under generally warm periods ($SSTA^+$) and MHWs. a–c** Represent **a** the NPPA during the $SSTA+$ period, with contributions from **b** sea surface temperature (SST)-dependent processes, and **c** SST-independent processes. **d–f** show the same decomposition but during MHW periods. **g–i** Present the difference in NPPA and its components between MHW and $SSTA^+$ periods. **j** Summarizes regional mean NPPA and its components across major global oceanic regimes, including the global ice-covered region, western boundary currents and their extensions (WBCEs), EBUCs, central-to-eastern tropical Pacific (CETP), the Southern Ocean, and subtropical gyre interiors (SGIs) (definitions in

Fig. S6). Boxes denote the interquartile range (Q1–Q3), the horizontal line indicates the median, and whiskers extend to 1.5× the interquartile range. Black points mark the mean value. Colored bars indicate total NPPA (blue), SST-dependent contributions (red), and SST-independent contributions (yellow), shown for both $SSTA^+$ (lighter shades, left) and MHW (darker shades, right) conditions. **k** Analogous to **j** but presents NPPA and its components across the 45 LMEs aggregated globally and by continent (grouping details in Fig. S7 and Table S1). The same box-and-whisker and color conventions apply.

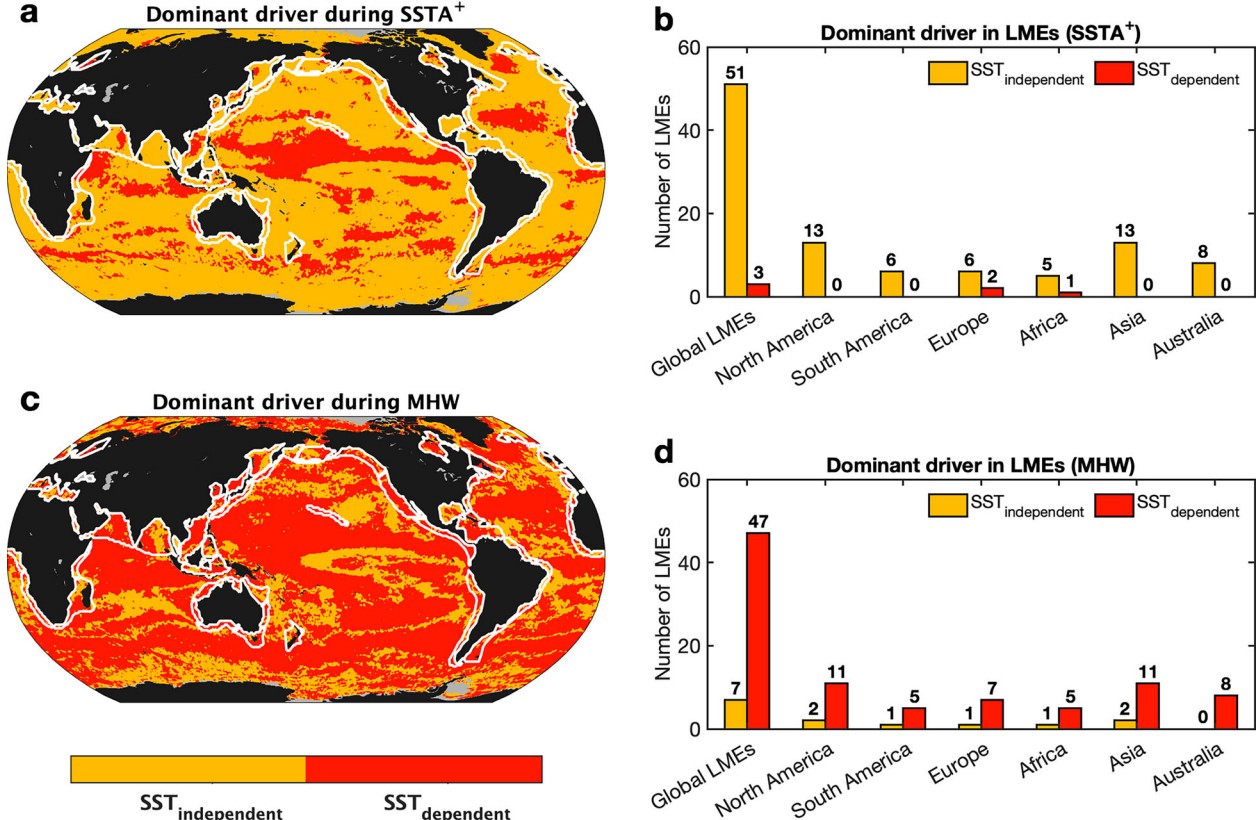

**Fig. 3 | Spatial and regional primary contributors of oceanic NPPA under generally warm periods ($SSTA^+$) and MHWs. a** Spatial distribution of the dominant contributor to NPPA during $SSTA^+$ periods, classified as either sea surface temperature (SST)-dependent or SST-independent. **c** as in **a**, but for MHWs. Regions shaded in yellow indicate NPPA is primarily driven by SST-independent processes, whereas red regions indicate dominance by SST-dependent processes.

White lines delineate boundaries of LMEs. Bar charts show the number of LMEs in which NPPA is primarily influenced by SST-dependent versus SST-independent processes, summarized by continent **b** during $SSTA^+$ periods and **d** during MHWs, grouped by continent. The area ratio of SST-dependent processes dominant during the $SSTA^+$ periods and MHWs are in Table S2.

MHW-inhibited. Applying the same approach to the 54 LMEs yields a similar distribution, with 35 LMEs classified as MHW-enhanced and 19 as MHW-inhibited, largely following latitudinal gradients.

Baseline environmental states differ markedly between the two groups. MHW-enhanced regions generally exhibit deeper mixed layers, cooler surface temperatures, lower climatological PAR, and higher background chlorophyll and nutrient concentrations (Fig. 4b, d, f, h, j, l, n). In contrast, MHW-inhibited regions are characterized by shallower mixed layers, warmer SST, higher baseline PAR, and lower nutrient and chlorophyll levels (Fig. 4c, e, g, i, k, m, o). Climatological mixed layer depth (MLD) exceeds 100 m across large portions of the North Atlantic and Southern Ocean (Fig. S8a), whereas shallow mixed layers dominate subtropical gyres.

Consistent with previous work[35], MHWs are accompanied by mixed-layer shoaling in both categories. The interquartile range (IQR) of MLD anomalies spans 0–9 m in MHW-enhanced regions and 0–5 m in MHW-inhibited regions (Figs. 4e and S8c). Despite this shoaling, nutrient anomalies during MHWs remain within their respective baseline ranges and decline in both categories (Fig. 4k, m, o). The amplitude of nutrient reduction is often larger in MHW-enhanced regions, but these regions begin from substantially higher nutrient baselines (Fig. 4j, l, n), implying that even moderate declines may still leave sufficient nutrient inventories to support productivity.

Chlorophyll and PAR exhibit contrasting responses between categories. Baseline CHL is substantially higher and more variable in MHW-enhanced regions than in inhibited regions (Fig. 4f). During MHWs, CHL anomalies tend to be positive in enhanced regions but negative in inhibited regions (Fig. 4g). PARA display a similar contrast:

although climatological PAR is lower poleward, MHWs are associated with larger PAR increases in enhanced regions than in inhibited regions (Fig. 4h, i). These patterns suggest that the net biomass response to MHWs reflects the combined influence of baseline nutrient–light conditions and their coevolution during extreme warming, consistent with previous evidence that high-latitude and upwelling systems exhibit greater flexibility in balancing light and nutrient constraints[35].

## Discussion

Our results reveal a systematic shift in how much of the ocean productivity anomaly during MHWs is statistically aligned with the temperature anomaly. By partitioning NPPA, we find that the fraction of NPPA variance that is captured by a linear SSTA-associated component increases during MHWs (Fig. 3c). This pattern should not be interpreted as the mechanistic dominance of temperature over ecological pathways. Rather, it indicates that during MHWs, a larger share of productivity variability becomes linearly coherent with the warm-anomaly state, consistent with a "bundled" response in which elevated SST and SST-correlated physical and biological processes jointly shape NPPA. Marine ecosystems typically respond to warming through multiple interacting processes—including nutrient–light colimitation, photoacclimation, photoinhibition, chlorophyll loss, and changes in grazing or mortality[5,8,36]—which collectively weaken simple temperature–productivity covariances under moderate conditions. During MHWs, however, these response pathways appear to narrow, leading to an increased SST-coherent component of NPPA without implying a change in causal ordering or a single-factor temperature control.

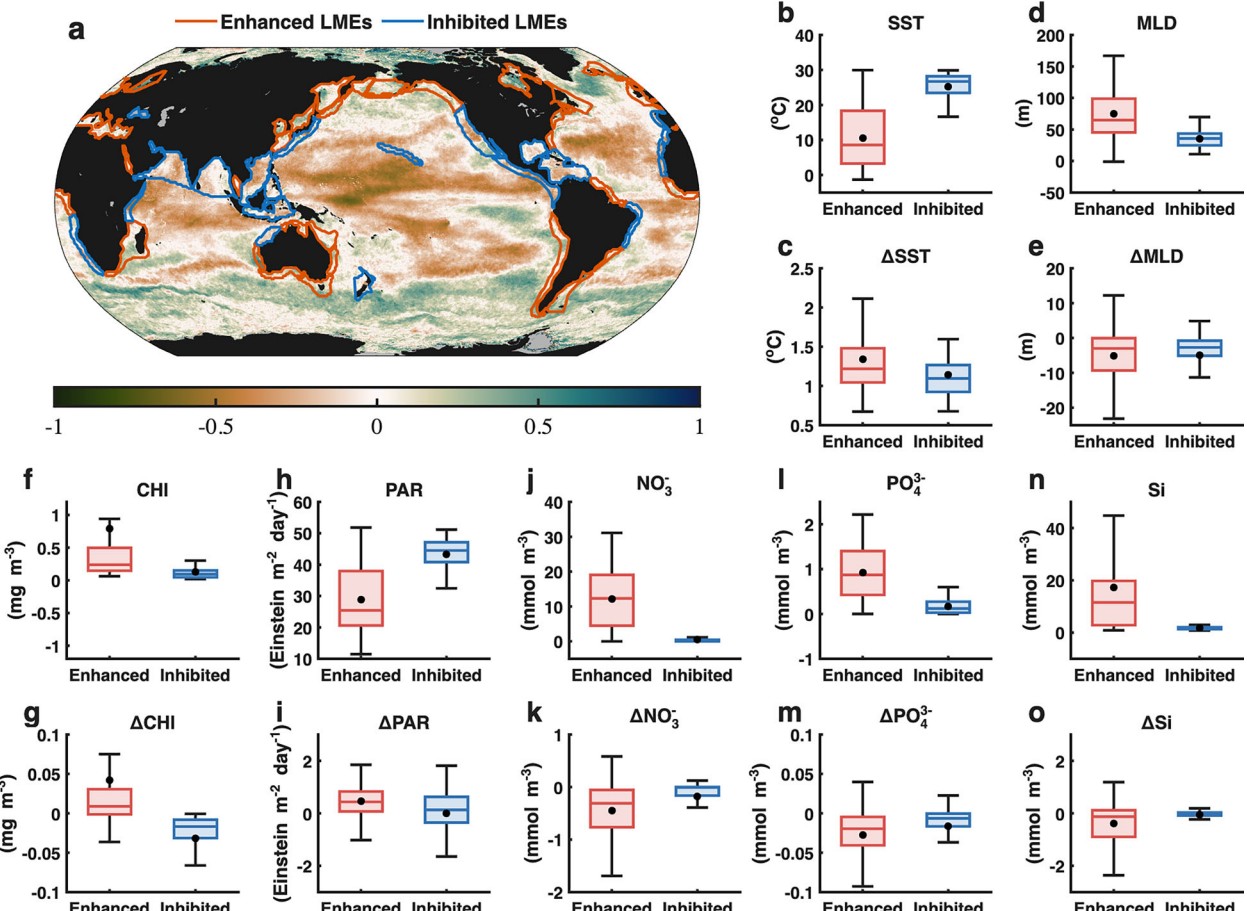

**Fig. 4 | Statistical contrasts between MHW-enhanced and MHW-inhibited regions. a** Spatial distribution of the correlation between oceanic NPPA and sea surface temperature anomaly (SSTA), with green shading indicating positive correlations and brown indicating negative correlations. Regions where NPPA–SSTA correlations are significantly positive (confidence level > 90%) are classified as MHW-enhanced regions, and significantly negative correlations define MHW-inhibited regions. LMEs are outlined and separated into MHW-enhanced LMEs (red boundaries) and MHW-inhibited LMEs (blue boundaries). **b–o** MHW-enhanced regions are labeled "Enhance" (red bars), and MHW-inhibited regions are labeled "Inhibit" (blue bars); black points denote group means, and whiskers show interquartile ranges. The mean state of **b** baseline SST and **c** SSTA during MHWs. The mean state of **d** baseline mixed layer depth (MLD) and **e** MLD anomalies during MHWs. The mean state of **f** baseline chlorophyll concentration (CHL) and **g** CHL anomalies during MHWs. The mean state of **h** baseline PAR and **i** PARA during MHWs. The mean state of **j** baseline nitrate ($NO_3^-$) concentrations and **k** $NO_3^-$ anomalies during MHWs. The mean state of **l** baseline phosphate ($PO_4^{3-}$) concentrations and **m** $PO_4^{3-}$ anomalies during MHWs. The mean state of **n** baseline silicate (Si) concentrations and **o** Si anomalies during MHWs.

The NPPA response to MHWs exhibits a pronounced geographical divide, with high-latitude regions generally showing productivity enhancement and low-latitude regions experiencing consistent declines (Fig. 1c). This contrast arises from the different baseline nutrient–light environments in which MHWs occur[37,38]. High-latitude systems begin from nutrient-rich, high-CHL conditions and are often limited by light; MHWs in these regions frequently coincide with persistent high-pressure anomalies[32,39] that increase surface irradiance, allowing NPPA to be maintained or even increased despite elevated temperatures. By contrast, subtropical and tropical oceans operate near the limit of nutrient availability. In these low-nutrient, high-light environments, additional stratification associated with MHWs reduces vertical nutrient supply, suppresses CHL, and leads to widespread productivity losses.

Classifying the global ocean into MHW-enhanced and MHW-inhibited regimes (Fig. 4) highlights a coherent spatial structure in ecosystem sensitivity. Nutrient-rich high latitudes and coastal upwelling regions show greater ecological buffering capacity, supported by deeper mixed layers, cooler background temperatures, and higher nutrient inventories. These conditions expand the viable ecological response space during extreme warming, allowing productivity to remain resilient. In contrast, nutrient-poor subtropical and tropical oceans—characterized by strong stratification and chronically low macronutrient supply—are consistently vulnerable to MHW-induced declines. This spatial differentiation suggests that the vulnerability of marine ecosystems to extreme warming depends not only on the magnitude of temperature anomalies but also on baseline ecosystem state and feedback mechanisms, including photoacclimation, nutrient–light colimitation, and grazing dynamics[38,40,41].

The increasing fraction of NPPA that covaries linearly with SSTA during MHWs raises concerns about how continued warming may alter the flexibility of marine ecosystems. As MHWs become more intense, frequent, and persistent[9], regions with limited buffering capacity may face growing exposure to direct thermal perturbations, with implications for biogeochemical cycling and ecosystem services. A key direction for future work is to resolve how different characteristics of MHWs—such as duration, cumulative intensity, depth penetration, and vertical structure—shape distinct ecological response pathways. Incorporating extreme-event dynamics into marine productivity projections will be essential for improving the representation of ecosystem sensitivity in Earth system models and for strengthening climate-resilient fisheries and ocean management strategies.

## Methods

### Data for the study

Satellite-derived estimates of monthly NPP, CHL, SST, and PAR, were obtained from the Ocean Color website (see section of "Data availability"), based on a variant of the Vertically Generalized Production Model (VGPM)[42]. The VGPM estimates depth-integrated NPP using surface CHL, SST, PAR, and day length. This study specifically focuses on the Eppley-VGPM version[34], which modifies the temperature-dependence of chlorophyll-specific NPP ($P^B_{opt}$)[31,43] by adopting an exponential increase with SST, in contrast to the standard VGPM's peak at 20 °C[44]. To evaluate the robustness of our findings, two additional global NPP products were analyzed: the carbon-based production model (CbPM)[28,29] and the CAFÉ model[27] (Fig. S3). SST is not an input factor in both the CAFÉ and CbPM NPP models, which helps to confirm that the MHW effect is not solely due to the parameterization of $P^B_{opt}$ as a function of temperature.

Monthly MLD data were derived from the Bluelink Reanalysis version 2020 (BRAN2020)[45], a high-resolution eddy-permitting ocean reanalysis with ~0.1° horizontal resolution and 51 vertical levels. BRAN2020 assimilates a broad suite of satellite and in situ observations—including altimetry, SST, Argo profiles, and surface salinity—into an updated global configuration of the Ocean Forecasting Australia Model. Its fine spatial and vertical resolution has demonstrated skill in representing upper-ocean stratification and mesoscale variability[46-48], both of which strongly influence nutrient distributions and mixed-layer light conditions during thermal extremes.

Nutrient concentrations used in Fig. 4 were obtained from the Global Ocean Biogeochemistry Non-Assimilative Hindcast product[49] provided by the Copernicus Marine Environment Monitoring Service (CMEMS). This global simulation (1998–2018) couples the NEMO physical ocean model with the PISCES biogeochemical module, providing internally consistent nitrate, phosphate, silicate, and iron fields that allow interpretation of nutrient-related constraints on productivity.

### Definition of MHWs and $SSTA^+$

MHWs are defined following Hobday et al.[40] as periods during which SST exceeds the seasonally varying 90th percentile threshold. For consistency with the satellite-based NPP record and to maintain a robust baseline, monthly MHWs during 1998–2018 were defined using a fixed 1998–2021 climatology[4,8]. Although mean SST exhibits long-term warming, the 90th-percentile threshold remains sufficiently above the evolving mean such that MHWs during this period constitute discrete warm extremes rather than a shift in baseline conditions.

To distinguish MHWs from more general warm anomalies, we define general $SSTA^+$ conditions as periods where SSTA is positive but remains below the MHW threshold. In other words, $SSTA^+$ conditions represent moderate warming events, while MHWs denote statistically rare extreme thermal anomalies. These two categories allow comparison of ecosystem responses under moderate versus extreme thermal forcing.

### Definition of LMEs

LMEs are expansive ocean regions (≥200,000 km²) that extend from coastal zones, including river basins and estuaries, to continental shelf breaks or major ocean currents just off the shelf. They are defined based on four criteria, including bathymetry, hydrography, productivity, and trophically linked populations[25]. LMEs play a key role in global ocean productivity, supporting rich biodiversity. LMEs occupy 22% of the ocean's surface but contribute to 95% of the global fish catch, providing goods and services to billions of people worth more than US\$12.6 trillion annually[15]. Based on the distribution of oceanic fisheries and dataset availability, we restrict our analysis to LMEs within 80° N/S and classify them by continent following the classification in Guo et al.[24] (Fig. S7 and Table S1).

### Statistical modeling of NPPA

Because the VGPM formulation multiplies several environmental terms, isolating the contribution of each factor to NPP variability is nontrivial. To quantify how thermal forcing and other processes jointly regulate NPPA, we construct a multivariate linear regression framework that partitions CHLA and PARA anomalies into components linearly associated with SSTA anomalies and components unexplained by SSTA.

Firstly, the NPP factors of CHL anomaly (CHLA) and PAR anomaly (PARA) were regressed onto the SSTA-related components ($CHLA_{sst}$ and $PARA_{sst}$) and SSTA-unrelated components ($CHLA_{res}$ and $PARA_{res}$) as shown in Eq. (1):

$$\text{CHLA} = \underbrace{\alpha\text{SSTA}}_{CHLA_{sst}} + CHLA_{res}, \quad \text{PARA} = \underbrace{\beta\text{SSTA}}_{PARA_{sst}} + PARA_{res}, \tag{1}$$

NPPA was then further regressed as

$$\text{NPPA} = \underbrace{a\text{SSTA} + bCHLA_{sst} + cPARA_{sst}}_{SST-\text{dependent NPPA}} + \underbrace{dCHLA_{res} + ePARA_{res} + NPPA_{res}}_{SST-\text{independent NPPA}}. \tag{2}$$

The first three terms on the right side of Eq. (2) are categorized into "SST-dependent NPPA", representing NPPA components that are linearly related to SSTA, $CHLA_{sst}$, and $PARA_{sst}$. These terms quantify NPP variability directly attributable to thermal effects. The sum of the remaining three terms on the right side are referred to as "SST-independent NPPA", including the linear responses of NPPA to $CHLA_{res}$ and $PARA_{res}$, and residual ($NPPA_{res}$). $NPPA_{res}$ encompasses all unexplained influences, such as shifts in phytoplankton community structure, modulation of trace elements affecting marine productivity, and nonlinear interactions between SSTA and NPPA. The evaluation and the coefficients of each term are shown in Fig. S9, and contributions of all six terms are individually quantified in Fig. S5.

## Data availability

All datasets used in this study are publicly accessible. NPP (Eppley-VGPM, CbPM, CAFÉ), CHL, SST, and PAR data from 1998–2021 are available from the Oregon State University Ocean Productivity website (http://orca.science.oregonstate.edu/npp_products.php). BRAN2020 mixed-layer depth fields are provided by Australia's National Computational Infrastructure[45], and the BRAN2020 dataset is accessible at https://geonetwork.nci.org.au/geonetwork/srv/eng/catalog.search#/metadata/f9372_7752_2015_3718. Nutrient concentrations were obtained from the CMEMS global multiyear biogeochemical hindcast product[49], available at https://data.marine.copernicus.eu/product/GLOBAL_MULTIYEAR_BGC_001_029/description.

## Code availability

Matlab2024b is used for plotting. The MATLAB code to identify MHWs is available at: https://github.com/ZijieZhaoMMHW/m_mhw1.0. The regression framework used to quantify NPPA responses to MHWs can be obtained from https://github.com/ZijieZhaoMMHW/MHW_NPP. The scripts and accompanying example data are available via Zenodo[54]. Coastlines and land boundaries for map visualizations were generated using MATLAB m_map with the GSHHG coastline database (https://www.aoml.noaa.gov/ftp/phod/hooper/CTDCAL/Jay_toolbox/m_map/map.html).

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

## Acknowledgements

This research was supported by the Australian Research Council (ARC) Centre of Excellence for Climate Extremes (CLEX; CE170100023). N.J.H. also acknowledges support from the ARC Centre of Excellence for the Weather of the 21st Century (CE230100012). Computing resources were provided by the National Computational Infrastructure (NCI), which is supported by the Australian Government. CLEX also sponsored the collaborative visit of C.B. to the Institute for Marine and Antarctic Studies, University of Tasmania, in 2024. C.B. was also supported by the U.S. National Science Foundation (NSF) through the ROCCA project (OCE –2400433). We thank the NASA Ocean Color website for providing access to the high-quality NPP datasets that made this study possible.

## Author contributions

C.B. conceived the study and led the analysis and manuscript writing. Z.Z. led the modeling and computations. N.J.H. and P.G.S. contributed to interpreting the results and revising drafts of this manuscript. L.W. provided early-stage supervision and contributed to the final revision of the manuscript. All authors discussed the results and approved the final manuscript.

## Competing interests

The authors declare no competing interests.
