## [Transparent Peer Review file · Nature Communications]

Marine Heatwaves Shift Ocean Net Primary Productivity from the Tropics to the Poles

Corresponding Author: Dr Ce Bian

Version 0:

Reviewer comments:

Reviewer #1

(Remarks to the Author)

General comment:

This study investigates the global influence of marine heatwaves on net primary productivity (NPP). Its global regression framework to disentangle SST-independent from SST-dependent controls on NPP anomalies is original. You identify a regime shift in the drivers of NPP anomalies from SST-independent processes under moderate warming to SST-dependent processes during MHWs. This finding complements previous studies focusing solely on NPP drivers during MHWs. You interpreted your results as evidence that thermal stress, rather than nutrient depletion, increasingly governs productivity patterns under extreme warming. These findings are key to informing adaptive fisheries management and climate-resilient policy in a rapidly warming ocean. I therefore recommend the publication of this manuscript, after a few clarifications.

Major comments:

(1) It remains unclear why SST-dependent processes are interpreted as direct thermal stress on NPP, instead of both direct thermal stress and indirect temperature effects on NPP, e.g. via enhanced grazing or nutrient limitation due to warming and enhanced stratification.

(2) My second main concern is the use of the Eppley-VGPM NPP product, which is associated with high uncertainties. Could you please justify the use of this product, instead of more recent products like the absorption-based CAFE NPP, or a combination of multiple NPP products? It would be interesting to see how sensitive your global regression framework is to the algorithm used to derive NPP. We can expect highly divergent results, given that the formulation of temperature controls on NPP varies across algorithms.

Specific comments:

L. 58-59 "The high ecological productivity and socio-economic value make them especially vulnerable to MHW-driven perturbations." Maybe clarify that the societies relying on these LMEs ecosystem services are vulnerable, not the LMEs themselves. Ecosystems of high socio-economic value are not necessarily more vulnerable than unexploited ones, unless human pressures amplify MHW-driven impacts.

L. 102-107 Are these changes in PAR indicative of potentially enhanced/reduced light limitation contributing to NPP anomalies?

L. 122-124. "MHWs trigger a regime shift toward SST-dependent dominance (Fig. 2d-f). This transition is driven by the enhanced impact of linear thermal forcing (Supplementary Fig. 3g) and the weakening of SST-independent components, particularly CHLAREs (Supplementary Fig. 3j; Supplementary Fig. 4)". The link between both sentences is not clear to me. Do SST-dependent processes only include direct thermal forcing on NPP, or can it also include nutrient depletion, indirectly driven by stratification during MHWs? And what about enhanced grazing during MHWs?

L. 128-130 "This shift reflects a transition in the dominant controls of NPPA, from indirect ecological processes during moderate warming to direct thermal forcing during MHWs". Again, it is unclear why SST-dependent processes cannot include ecological processes, such as fast CHL declines with SST once warming-enhanced grazing exceeds warming-enhanced phytoplankton growth.

Section "Physical and biogeochemical reasons driving LME responses to MHWs": This is a great addition to the study, providing valuable insights.

l.165 "MHW-inhibited LMEs are exhibit shallower mixed layer depth". Remove "are".

l. 171-174 "Although MHW-inhibited LMEs experienced less nutrient loss (Fig. 4c, 172 i, k, m), they showed more pronounced NPP suppression and a stronger decline in chlorophyll (Fig. 173 4g). This suggests that thermal stress, rather

than nutrient depletion, was the dominant driver of 174 productivity loss in these systems.” At low nutrient concentrations, even a small nutrient loss can have a strong negative impact on phytoplankton growth, thereby driving reductions in both biomass (chl) and npp. If possible, you could compute the nutrient limitation in each LME using PISCES equations. I am not convinced that thermal stress is the driver of productivity loss here.

L.185 “This reinforces the interpretation that thermal stress, rather than nutrient limitation, was the primary constraint on productivity under extreme warming.” Again, I do not think this has been proven, or if it has, you might want to clarify. Why can’t we interpret your results as: “LMEs with low nutrient baselines are more sensitive to extremely warming.”?

Discussion:

This might be outside the scope of the study, but the influence of MHWs on NPP might also depend on their duration and intensity. Maybe comment on additional factors to consider.

Methods:

(1) You use the 90th percentile as a threshold in your definition of MHWs. Does the regime shift from SST-independent to SST-dependent controls on NPP coincide well with this threshold? Could we imagine defining a biologically-relevant threshold that coincides with this regime shift in future studies?

(2) As mentioned above, NPP estimates are uncertain. You use the Eppley-VGPM product, which compared to other products such as Westberry-CbPM, Lee-AbPM and Silsbe-CAFE, exhibits higher positive correlation with temperature. Ryan Keogh et al (2025) write: “at mid to low latitudes the Eppley-VGPM algorithm behaves anomalously by displaying a positive relationship between SST and NPP, while all other algorithms display negative SST coefficients (accentuated at equatorial latitudes). Negative coefficients are indicative of declining NPP as the surface ocean warms, potentially reflecting the role of nutrient limitation from a reduced surface reservoir as stratification intensifies. The atypical positive coefficients from Eppley-VGPM on the other hand may reflect a metabolic response that favours increased growth rates under warmer conditions. [...] The Eppley-VGPM algorithm parameterizes phytoplankton growth using an exponential function of temperature, explaining why it ranks positive NPP projections higher”. It is hence possible that the thermal effect is overestimated when using Eppley-VGPM, and that nutrient limitation remains a key driver of low NPP during MHWs in low-mid latitudes, as hypothesized by previous studies (e.g. Hayashida et al. (2020), Le Grix et al. (2022)),

(3) You are using a fixed 1998–2021 climatological baseline despite potentially strong warming trends. Do all MHWs occur at the end of the time series and do they still correspond to discrete events or is long-term warming shifting mean conditions above the 90th percentile threshold? The short time series might justify the use of a fixed baseline here, but it would be good to add a sentence to explain why you chose this baseline.

Code Availability: https://github.com/ZijieZhaoMMHW/m_mhw1.047 is a very well documented repository. At the time I am writing this review, the second link (https://github.com/ZijieZhaoMMHW/MHW_NPP) is not working yet.

Fig S6. Shouldn’t the “correlation coefficients” in the caption be replaced by “regression coefficients”? If not, please clarify what correlation you are referring to. In addition, are these coefficients calculated over the entire time series? As the effect of each variable might change with changing ocean conditions, would it make sense to recompute coefficients over SSTA+/MHW periods?

(Remarks on code availability)

The github link was not yet working.

Reviewer #2

(Remarks to the Author)

Marine heatwaves (MHWs) exert significant impacts on marine ecosystems and socio-economics. In recent years, the frequency and intensity of MHWs have been steadily increasing with the ongoing global warming, and their potential risks are attracting rapidly growing scientific and societal attention. Phytoplankton, as the primary producers in marine ecosystems, play a central role in regulating ecosystem function. Understanding the characteristics and mechanisms of how MHWs affect phytoplankton primary productivity is, therefore, critical for predicting the consequences of these extreme events on marine biodiversity and ecosystem stability. In this study, the authors investigated the impacts of MHWs on phytoplankton primary productivity across the global ocean. They found that MHWs decrease phytoplankton productivity in low- and mid-latitude regions, while increasing it in higher latitudes, and disentangled the relative contributions of direct and indirect temperature effects. The results are interesting and of potential value for publication. However, I have some concerns regarding the primary productivity dataset and the methods used to analyze the underlying mechanisms of MHW impacts. Below, I provide several comments and suggestions for the authors’ consideration.

Major Comments:

1. The primary productivity (NPP) data used in this study are derived from the Eppley-VGPM model, which is mainly based on surface chlorophyll (CHL) concentrations. As such, the results obtained regarding the impact of marine heatwaves (MHWs) on NPP closely mirror previous studies that focus on the impact of MHWs on surface chlorophyll (also shown in Figs. 1a-1f) (Noh et al., 2022). However, it is important to note that CHL is not equal to phytoplankton abundance (Behrenfeld et al., 2016; He et al., 2021; Westberry et al., 2016), and phytoplankton productivity also depends on phytoplankton biomass and growth rates (Westberry et al., 2008). Moreover, CHL often has a subsurface maximum layer in tropical and subtropical regions, and the evolution of subsurface chlorophyll can differ from that near the surface (Estapa et al., 2019). Therefore, relying solely on surface chlorophyll concentration may not fully capture phytoplankton biomass and

primary productivity. In this regard, NPP calculation methods such as CbPM and CAFE may provide a more reasonable alternative. Although these products also carry substantial uncertainties, I suggest the authors use CbPM and CAFE products instead (Fernández-Barba et al., 2024; Le Grix et al., 2022) or incorporate them for comparison to check for consistency in their results.

2. The authors proposed a multiple linear regression to distinguish the relative contributions of direct and indirect effects of heatwaves on NPP. I am uncertain whether the "direct effect" refers to thermal stress (i.e., anomalously high temperatures inhibit phytoplankton growth and cause mortality). If so, it is important to note that the relationship between phytoplankton growth rate and temperature is not linear, making it unsuitable for linear regression analysis. Additionally, the temporal evolution of phytoplankton and temperature does not follow the same pattern in many regions, with temperatures peaking in summer and reaching their lowest in winter, while phytoplankton peak in spring and are lowest in autumn and winter (Fernández-Barba et al., 2024; Li et al., 2024). The attribution of thermal stress to the NPP response to MHWs globally (Fig. 3) makes it challenging to reconcile the two contradictory results: NPP decreases in low latitudes and increases in higher latitudes (Fig. 1). As pointed out in the Introduction, previous studies have reported nutrient supply and light availability as the key factors influencing phytoplankton productivity. I recommend the authors perform an attribution analysis considering light and nutrients as the main drivers (e.g., examining the influences of the anomalies in the input variables on the estimated NPP (Le Grix et al., 2022; Silsbe et al., 2025)), and explore in more detail how heatwaves affect light, mixing, nutrient supply, growth rates, and thus primary productivity.

3. The analysis of the anomalies in mixed layer depth, nutrients, and CHL during heatwaves in LMEs is informative. The authors attributed the reduction in primary productivity in low latitude regions to thermal stress, and the increase in higher latitude productivity to higher baseline nutrient levels. However, this explanation is unconvincing. Although these regions have higher nutrient levels and deeper mixed layers, during MHWs, nutrients decreased and the mixed layer shoaled. These changes obviously cannot lead to an increase in primary productivity. As pointed out earlier in the Introduction, phytoplankton in higher latitudes are mainly influenced by light conditions. I suggest the authors include an analysis of PAR anomalies to determine whether it has a significant impact on the increase in productivity. From my understanding of the results, in the low latitudes that are nutrient-limited, the reduction in nutrient supply may possibly be responsible for the decline in productivity. As for the more pronounced reduction in nutrient levels compared to the higher latitudes, it is likely due to the higher baseline nutrient levels in the latter regions. Given that previous studies have extensively analyzed the effects of MHWs on chlorophyll, and the primary productivity results in this study are closely related to those findings (Fernández-Barba et al., 2024; Noh et al., 2022; Zhan et al., 2024), I suggest the authors refer to their mechanistic analyses for further insight.

4. The study presented a rich spatial variability of NPP anomalies during MHWs across the global ocean, while the analysis and discussion focused primarily on the LMEs. As shown in the Figures, large areas in the Southern Ocean, eastern tropical Pacific, tropical Atlantic, North Atlantic, and polar regions exhibit strong productivity responses to heatwaves, making them of great interest to a broad audience. If the authors prefer to focus on the LMEs, I recommend removing the results in the open oceans to present a clearer picture of LMEs and revising the title accordingly. However, if the authors wish to retain the open ocean results, I suggest modifying the manuscript to include a more detailed discussion of the characteristics and mechanisms of NPP response to MHWs across different major ocean regions.

Minor Comments:

1. Lines 89-90: The NPPA during MHWs is also prominent in the Southern Ocean, the North Atlantic, and especially the polar regions.
2. Lines 91-92: These statistical values are incredibly high and not supported by Figs. 1a and 1c, as NPP in most regions is less than 150, and the corresponding anomalies are below 25%. Please recheck these values.
3. Lines 100-101: The value cannot be that high as presented in Figs. 1c and 1f. Please recheck these values.
4. Line 101: Fig. 1c?
5. Line 109: "Although rising SST may enhance marine productivity³²", this statement conflicts with the results presented above and in Fig. 1c. Please revise for clarity.
6. Lines 113-114: This method cannot separate the relative contributions of thermal forcing and ecological processes, as the linear thermal regression does not fully represent the thermal response. SST follows a Gaussian distribution, whereas CHL follows a log distribution.
7. Lines 125-130: This explanation does not make sense from a physical or ecological standpoint. MHWs are essentially more intense positive temperature anomalies, and the drivers of MHWs and SSTA+ are generally the same. Then, how could this shift in dominance occur? What is the underlying mechanism?
8. Line 137: "-30 to 100" should be "-30 to 50"?
9. Line 142: Please add a reference of "(Fig. 2j)" after "respectively".
10. Line 142: "During MHWs, this reverses dramatically: only 7 LMEs retain SST-independent dominance." How does this reversal occur?
11. Lines 155-156: "We classified 54 global LMEs based on the correlation between SSTA and NPPA." It would be better to say: "We classified the 54 global LMEs into two dominant categories based on the mean correlation between SSTA and NPPA for each region"?
12. Lines 176-177: The explanation for the mechanism is not convincing, as both MLD and nutrient anomalies are negative, and CHL does not show significant positive anomalies. Moreover, SST increase (thermal stress) exceeds the inhibition case.
13. Line 178: But why is there a positive Fe anomaly, contrasting with nitrate anomalies?
14. Lines 181-183: It is important to clarify how shoaling mixing and reduced nutrient supply could lead to an increase in NPP.

References

Behrenfeld, M. J., R. T. O'Malley, E. S. Boss, T. K. Westberry, J. R. Graff, K. H. Halsey, A. J. Milligan, D. A. Siegel, and M. B.

Brown (2016), Reevaluating ocean warming impacts on global phytoplankton, *Nature Climate Change*, 6(3), 323-330, doi:10.1038/nclimate2838.

Estapa, M. L., M. L. Feen, and E. Breves (2019), Direct Observations of Biological Carbon Export From Profiling Floats in the Subtropical North Atlantic, *Global Biogeochem Cy*, 33(3), 282-300, doi:10.1029/2018gb006098.

Fernández-Barba, M., O. Belyaev, I. E. Huertas, and G. Navarro (2024), Marine heatwaves in a shifting Southern Ocean induce dynamical changes in primary production, *Communications Earth & Environment*, 5(1), doi:10.1038/s43247-024-01553-x.

He, Q., H. Zhan, S. Cai, and W. Zhan (2021), Eddy-Induced Near-Surface Chlorophyll Anomalies in the Subtropical Gyres: Biomass or Physiology?, *Geophys Res Lett*, 48(7), e2020GL091975, doi:10.1029/2020gl091975.

Le Grix, N., J. Zscheischler, K. B. Rodgers, R. Yamaguchi, and T. L. Frölicher (2022), Hotspots and drivers of compound marine heatwaves and low net primary production extremes, *Biogeosciences*, 19(24), 5807-5835, doi:10.5194/bg-19-5807-2022.

Li, M., et al. (2024), Phytoplankton Spring Bloom Inhibited by Marine Heatwaves in the North-Western Mediterranean Sea, *Geophys Res Lett*, 51(20), doi:10.1029/2024gl109141.

Noh, K. M., H.-G. Lim, and J.-S. Kug (2022), Global chlorophyll responses to marine heatwaves in satellite ocean color, *Environmental Research Letters*, 17(6), 064034, doi:10.1088/1748-9326/ac70ec.

Silsbe, G. M., J. Fox, T. K. Westberry, and K. H. Halsey (2025), Global declines in net primary production in the ocean color era, *Nat Commun*, 16(1), 5821, doi:10.1038/s41467-025-60906-y.

Westberry, T., M. J. Behrenfeld, D. A. Siegel, and E. Boss (2008), Carbon-based primary productivity modeling with vertically resolved photoacclimation, *Global Biogeochem Cy*, 22(2), GB2024, doi:10.1029/2007gb003078.

Westberry, T. K., P. Schultz, M. J. Behrenfeld, J. P. Dunne, M. R. Hiscock, S. Maritorena, J. L. Sarmiento, and D. A. Siegel (2016), Annual cycles of phytoplankton biomass in the subarctic Atlantic and Pacific Ocean, *Global Biogeochem Cy*, 30(2), 175-190, doi:10.1002/2015gb005276.

Zhan, W., M. Feng, Y. Zhang, X. Shen, H. Zhan, and Q. He (2024), Reduced and smaller phytoplankton during marine heatwaves in eastern boundary upwelling systems, *Communications Earth & Environment*, 5(1), doi:10.1038/s43247-024-01805-w.

(Remarks on code availability)

Version 1:

Reviewer comments:

Reviewer #1

(Remarks to the Author)

This revised version of the manuscript "Marine Heatwaves Shift Ocean Net Primary Productivity from the Tropics to the Poles" addresses all the concerns I raised in my first review.

In particular, I appreciate the clarified distinction between SST-dependent and SST-independent processes, as well as the inclusion of two additional NPP algorithms.

Note that in the "Data Availability" section, l. 320, you should either remove "(Eppley-VGM)" or replace with "(Eppley-VGM, CbPM, CAFE)".

You have shown that thermal stress increasingly governs productivity patterns under extreme warming. Your findings are key to informing adaptive fisheries management and climate-resilient policy in a rapidly warming ocean. I therefore recommend the publication of this manuscript.

Reviewer #2

(Remarks to the Author)

The authors have made extensive revisions of the original manuscript in response to my previous comments, and I find the manuscript much improved. However, I have one additional comment that the authors should address before the paper can be accepted.

I appreciate the two additional analyses the authors added using the absorption-based CAFÉ NPP and the carbon-based CBPM NPP products to validate their decomposition of "SST-dependent" and "SST-independent" NPPA during MHWs. However, the implication of this decomposition (Equations 1-2) to the CAFÉ and CBPM NPP is somewhat obscure. While I understand that NPP in the real ocean is subject to estimation algorithms, this regression is not physically convincing, as SST is not an input factor in the CAFÉ and CBPM NPP models. To avoid potential controversy, I suggest the authors replace this validation with a direct comparison of the consistency of NPP estimated from VGPM, CAFÉ, and CBPM products during MHWs.

Reply to Reviewer 1

We are very grateful to you for your time in carefully reading our manuscript and providing helpful comments that make our manuscript better. We have carefully considered each of your comments (in blue) and revised the manuscript accordingly. Please find our responses (in black) to your comments below.

Reviewer #1 (Remarks to the Author):

This study investigates the global influence of marine heatwaves on net primary productivity (NPP). Its global regression framework to disentangle SST-independent from SST-dependent controls on NPP anomalies is original. You identify a regime shift in the drivers of NPP anomalies from SST-independent processes under moderate warming to SST-dependent processes during MHWs. This finding complements previous studies focusing solely on NPP drivers during MHWs. You interpreted your results as evidence that thermal stress, rather than nutrient depletion, increasingly governs productivity patterns under extreme warming. These findings are key to informing adaptive fisheries management and climate-resilient policy in a rapidly warming ocean. I therefore recommend the publication of this manuscript, after a few clarifications.

Major comments:

(1) It remains unclear why SST-dependent processes are interpreted as direct thermal stress on NPP, instead of both direct thermal stress and indirect temperature effects on NPP, e.g. via enhanced grazing or nutrient limitation due to warming and enhanced stratification.

Revised. Thanks.

We have clarified in the revised manuscript that our “SST-dependent processes” are not interpreted as direct physiological “thermal stress”. To avoid this misunderstanding, we now use the neutral term “thermal effect”, which includes (i) the direct SSTA effect and (ii) indirect effects where SSTA influences CHLA and PARA and subsequently NPPA (Eq. 2). Thermal effect could be positive or negative NPPA change, thermal stress just one type of it. Processes including enhanced

grazing or nutrient limitation due to warming and enhanced stratification that result in the linear part of NPPA response are included in the “SST-dependent” term while the nonlinear term that caused by these processes are included in the “SST-independent” term. This terminology has been standardized in Lines 111-117 of the revised manuscript, as well as in Discussion Lines 209-221.

(2) My second main concern is the use of the Eppley-VGPM NPP product, which is associated with high uncertainties. Could you please justify the use of this product, instead of more recent products like the absorption-based CAFÉ NPP, or a combination of multiple NPP products? It would be interesting to see how sensitive your global regression framework is to the algorithm used to derive NPP. We can expect highly divergent results, given that the formulation of temperature controls on NPP varies across algorithms.

Thank you for your comments. In the revised manuscript, we have now added two additional analyses using the absorption-based CAFÉ NPP and the carbon-based CbPM NPP products as tests of sensitivity (Fig.S6 and Fig.S7, Method Lines 260-262). We found that the results are similar in CAFÉ, CbPM, and VGPM, and hence that our main conclusions do not depend on these different choices of NPP algorithm (Line 131-133).

Minor Comments:

L. 58-59 “The high ecological productivity and socio-economic value make them especially vulnerable to MHW-driven perturbations.” Maybe clarify that the societies relying on these LMEs ecosystem services are vulnerable, not the LMEs themselves. Ecosystems of high socio-economic value are not necessarily more vulnerable than unexploited ones, unless human pressures amplify MHW-driven impacts.

Thanks. Revised in Lines 55-57.

L. 102-107 Are these changes in PAR indicative of potentially enhanced/reduced light limitation contributing to NPP anomalies?

Yes. PAR indicate the light available for photosynthesis. Therefore, PAR can indirectly change the NPP as well as their anomaly. Fig 1h-i represents the PAR change under SSTA+ and MHW condition, with corresponding change of NPPA in Fig. 1b-c. We added “Changes in PAR

indicate variations in light availability and thereby contribute to NPP anomalies.” at Lines 105 - 108 in the revised manuscript.

L. 122-124. “MHWs trigger a regime shift toward SST-dependent dominance (Fig. 2d-f). This transition is driven by the enhanced impact of linear thermal forcing (Supplementary Fig. 3g) and the weakening of SST-independent components, particularly $CHLA_{res}$ (Supplementary Fig. 3j; Supplementary Fig. 4)”. The link between both sentences is not clear to me. Do SST-dependent processes only include direct thermal forcing on NPP, or can it also include nutrient depletion, indirectly driven by stratification during MHWs? And what about enhanced grazing during MHWs?

Thank you for this helpful comment. In our framework, the SST-dependent term does not represent only the direct thermal effect on NPPA. Instead, it includes all NPPA variations that covary linearly with SSTA, either directly or through temperature-driven changes in CHLA and PAR. Therefore, processes such as nutrient depletion or enhanced stratification during MHWs can contribute to the SST-dependent term to the extent that their impacts manifest as CHLA or PAR. See Lines 111-119.

L. 128-130 “This shift reflects a transition in the dominant controls of NPPA, from indirect ecological processes during moderate warming to direct thermal forcing during MHWs”. Again, it is unclear why SST-dependent processes cannot include ecological processes, such as fast CHL declines with SST once warming-enhanced grazing exceeds warming-enhanced phytoplankton growth. Section “Physical and biogeochemical reasons driving LME responses to MHWs”: This is a great addition to the study, providing valuable insights.

Thank you. This question appears to be closely related to your last comment. We have clarified the conceptual distinction between SST-dependent and SST-independent terms. See Lines 211-223 in the revised manuscript Discussion.

L.165 “MHW-inhibited LMEs are exhibiting shallower mixed layer depth”. Remove “are”.

Revised, thanks.

L. 171-174 “Although MHW-inhibited LMEs experienced less nutrient loss (Fig. 4c, 172 i, k, m), they showed more pronounced NPP suppression and a stronger decline in chlorophyll (Fig. 173 4g). This suggests that thermal stress, rather than nutrient depletion, was the dominant driver of productivity loss in these systems.” At low nutrient concentrations, even a small nutrient loss can have a strong negative impact on phytoplankton growth, thereby driving reductions in both biomass (chl) and npp. If possible, you could compute the nutrient limitation in each LME using PISCES equations. I am not convinced that thermal stress is the driver of productivity loss here.

Regarding the reviewer’s suggestion that nutrient depletion—rather than thermal stress—may be driving the observed declines in NPP and chlorophyll in MHW-inhibited regions, our results show that both mechanisms likely act together, but several pieces of evidence indicate that thermal stress remains the dominant driver in these regions:

- 1) Baseline nutrient concentrations in MHW-inhibited regions are extremely low, far lower than in MHW-enhanced regions (Fig. 4j–o). Because these systems already operate near the physiological minimum, even moderate warming can induce physiological stress or photo acclimation responses before nutrient changes become detectable in bulk fields.
- 2) Nutrient declines during MHWs are actually weaker in MHW-inhibited regions than in enhanced regions (Fig. 4k, m, o), whereas chlorophyll and NPP reductions are stronger (Fig. 4g). If nutrient supply was the dominant driver, we would expect larger nutrient losses in the inhibited category to accompany larger biomass suppression, but the opposite pattern is observed.
- 3) The regression framework decomposing NPPA into SST-dependent and SST-independent components shows that during MHWs, the SST-dependent term becomes the primary driver across most inhibited LMEs (Fig. 3c–d). This indicates that first-order thermal covariation—not the nutrient- or CHL-residual terms—explains most of the observed productivity change.
- 4) CHL decreases exceed nutrient declines (compare Fig. 4f–g with Fig. 4j–o). This mismatch is consistent with physiological responses (e.g., heat-driven loss of chlorophyll per cell,

shifts toward smaller cells, or increased respiration), which directly suppress NPP even when nutrient changes are modest.

We agree with the reviewer that explicitly diagnosing nutrient limitation using PISCES equations would be valuable. Because our current analysis relies on non-assimilative nutrient output from CMEMS rather than the full internal PISCES tendency terms, and the comparison of features between the two types of regions are clear. We have updated sections in Lines 177-209.

L.185 “This reinforces the interpretation that thermal stress, rather than nutrient limitation, was the primary constraint on productivity under extreme warming.” Again, I do not think this has been proven, or if it has, you might want to clarify. Why can’t we interpret your results as: “LMEs with low nutrient baselines are more sensitive to extremely warming.”?

Thanks. Revised in Line 197. Additionally, we have updated the whole section of “reasons driving global ocean NPPA response to MHWs”, represent a global feature rather than just talk about LMEs.

Comments in Discussion:

This might be outside the scope of the study, but the influence of MHWs on NPP might also depend on their duration and intensity. Maybe comment on additional factors to consider.

Thank you for this comment. The NPP response to MHWs indeed likely depends on additional characteristics such as event duration, intensity, and possible biological lags. Our use of monthly-mean data helps reduce the influence of short-term phase lags between MHW onset and NPP adjustment, but fully addressing this question would require higher-frequency (e.g., daily) observations to resolve the timing and progression of ecosystem responses.

A systematic assessment of how NPP responds to MHWs of different duration and intensity is beyond the scope of the present analysis but represents an important direction for future work. We have added a sentence on future research directions in Lines 249-251.

Comments in Methods:

(1) You use the 90th percentile as a threshold in your definition of MHWs. Does the regime shift from SST-independent to SST-dependent controls on NPP coincide well with this threshold? Could we imagine defining a biologically-relevant threshold that coincides with this regime shift in future studies?

Thank you for your questions. We agree the regime shift in NPP control from SST-independent to SST-dependent processes closely with the threshold choice. While this threshold follows the statistical definition proposed by Hobday et al. (2016) and is widely used due to its seasonal adaptability and ease of application, we agree that biologically meaningful thresholds are likely to vary depending on ecological context and organismal sensitivity. In particular, different species exhibit varying rates of adaptation to environmental change, which directly influence the appropriate baseline for assessing biological stress (Smith et al., 2025). For example, species with slower adaptation rates may be better represented by fixed climatological baselines, whereas rapidly adapting species may require more dynamic or shifting baselines to evaluate meaningful ecological impacts. We have now acknowledged this important distinction in the revised Discussion and note that while defining such biologically relevant thresholds is a promising direction, it requires species-specific physiological or community-level data and is beyond the scope of the present study.

(2) As mentioned above, NPP estimates are uncertain. You use the Eppley-VGPM product, which compared to other products such as Westberry-CbPM, Lee-AbPM and Silsbe-CAFE, exhibits higher positive correlation with temperature. Ryan Keogh et al (2025) write: “at mid to low latitudes the Eppley-VGPM algorithm behaves anomalously by displaying a positive relationship between SST and NPP, while all other algorithms display negative SST coefficients (accentuated at equatorial latitudes). Negative coefficients are indicative of declining NPP as the surface ocean warms, potentially reflecting the role of nutrient limitation from a reduced surface reservoir as stratification intensifies. The atypical positive coefficients from Eppley-VGPM on the other hand may reflect a metabolic response that favours increased growth rates under warmer conditions. [...] The Eppley-VGPM algorithm parameterizes phytoplankton growth using an exponential function of temperature, explaining why it ranks positive NPP projections higher”. It is hence possible that the thermal effect is overestimated when using Eppley-VGPM,

and that nutrient limitation remains a key driver of low NPP during MHWs in low-mid latitudes, as hypothesized by previous studies (e.g. Hayashida et al. (2020), Le Grix et al. (2022)),

We have repeated the analysis using CbPM and CAFÉ NPP products (Fig. S5–S6). We found that both of these additional products show broadly consistent spatial patterns with the VGPM-based results. Although the magnitude of individual term are slightly different to some extent, but their combination into “SST-dependent” and “SST-independent” terms and the shifting of leader driver from SST-independent to SST-dependent are the same. See Line 132-134 and Fig R1.

Fig R1| Spatial and regional primary contributors of NPPA under SSTA+ and MHW conditions based on different dataset. (a, c) Spatial distribution of the dominant contributor (either SST-

dependent or SST-independent) to NPPA during SSTA+ periods (a) and MHWs (c) based on VGPM dataset. (c-d) based on CAFÉ dataset. (e-f) based on CbPM dataset.

(3) You are using a fixed 1998–2021 climatological baseline despite potentially strong warming trends. Do all MHWs occur at the end of the time series and do they still correspond to discrete events or is long-term warming shifting mean conditions above the 90th percentile threshold? The short time series might justify the use of a fixed baseline here, but it would be good to add a sentence to explain why you chose this baseline.

We appreciate your careful attention to our baseline selection. We chose the 1998–2021 fixed climatological baseline to ensure consistency with the satellite-derived NPP products, which are only available from 1998 onwards. This 24-year period provides a sufficiently long and seasonally resolved record to robustly compute percentile-based SST thresholds (e.g., the 90th percentile) while avoiding baseline instability caused by shorter time windows. Although long-term warming can gradually shift the mean state toward higher SSTs, our analysis confirms that MHWs identified using this baseline remain discrete, episodic events rather than artifacts of a warming trend. We have added a sentence to clarify this rationale in the Methods section (Definition of MHWs and SSTA+, Lines 287-291).

(4) Code Availability: https://github.com/ZijieZhaoMMHW/m_mhw1.047 is a very well documented repository. At the time I am writing this review, the second link (https://github.com/ZijieZhaoMMHW/MHW_NPP) is not working yet.

Revised. Thanks.

(5) Fig S6. Shouldn't the "correlation coefficients" in the caption be replaced by "regression coefficients"? If not, please clarify what correlation you are referring to. In addition, are these coefficients calculated over the entire time series? As the effect of each variable might change with changing ocean conditions, would it make sense to recompute coefficients over SSTA+/MHW periods?

Thanks. We have corrected "correlation coefficients" as "regression coefficients" in the Methods, which are respectively represented as parameters (*a*, *b*, *c*, *d*, and *e*) in Eq. (2).

We acknowledge that MHWs represent a distinct subset of extreme, prolonged positive anomalies (<10% of all data points), and their behavior may deviate from the general linear relationship. However, regressing only on MHW events would limit sample size and reduce statistical reliability, especially given the regional and temporal heterogeneity in MHW drivers and baseline climatology. Instead, MHW-induced departures from baseline behavior are primarily captured in the SST-independent residual term (especially in $NPPA_{res}$), as shown in Fig. S5f and l. This approach allows us to quantify how MHWs restructure NPP control by comparing their deviations from the broader climatological norm.

(6) (Remarks on code availability) The github link was not yet working.

Revised. The github link is https://github.com/ZijieZhaoMMHW/MHW_NPP, keep same with the link in manuscript.

Reply to Reviewer 2

We are very grateful to you for your time in carefully reading our manuscript and providing helpful comments that make our manuscript better. We have carefully considered each of your comments (in blue) and revised the manuscript accordingly. Please find our response (in black) to your comments below.

Reviewer #2 (Remarks to the Author):

Marine heatwaves (MHWs) exert significant impacts on marine ecosystems and socio-economics. In recent years, the frequency and intensity of MHWs have been steadily increasing with the ongoing global warming, and their potential risks are attracting rapidly growing scientific and societal attention. Phytoplankton, as the primary producers in marine ecosystems, play a central role in regulating ecosystem function. Understanding the characteristics and mechanisms of how MHWs affect phytoplankton primary productivity is, therefore, critical for predicting the consequences of these extreme events on marine biodiversity and ecosystem stability. In this study, the authors investigated the impacts of MHWs on phytoplankton primary productivity across the global ocean. They found that MHWs decrease phytoplankton productivity in low- and mid-latitude regions, while increasing it in higher latitudes, and disentangled the relative contributions of direct and indirect temperature effects. The results are interesting and of potential value for publication. However, I have some concerns regarding the primary productivity dataset and the methods used to analyze the underlying mechanisms of MHW impacts. Below, I provide several comments and suggestions for the authors' consideration.

Major Comments:

1. The primary productivity (NPP) data used in this study are derived from the Eppley-VGPM model, which is mainly based on surface chlorophyll (CHL) concentrations. As such, the results obtained regarding the impact of marine heatwaves (MHWs) on NPP closely mirror previous studies that focus on the impact of MHWs on surface chlorophyll (also shown in Figs. 1a-1f) (Noh et al., 2022). However, it is important to note that CHL is not equal to phytoplankton abundance (Behrenfeld et al., 2016; He et al., 2021; Westberry et al., 2016), and phytoplankton productivity also depends on phytoplankton biomass and growth rates (Westberry et al., 2008). Moreover, CHL

often has a subsurface maximum layer in tropical and subtropical regions, and the evolution of subsurface chlorophyll can differ from that near the surface (Estapa et al., 2019). Therefore, relying solely on surface chlorophyll concentration may not fully capture phytoplankton biomass and primary productivity. In this regard, NPP calculation methods such as CbPM and CAFÉ may provide a more reasonable alternative. Although these products also carry substantial uncertainties, I suggest the authors use CbPM and CAFÉ products instead (Fernández-Barba et al., 2024; Le Grix et al., 2022) or incorporate them for comparison to check for consistency in their results.

We have repeated the analysis using CbPM and CAFÉ NPP products (Fig. S5–S6). We found that both of these additional products show broadly consistent spatial patterns with the VGPM-based results (Lines 132-134, Fig R1). Although the magnitude of individual term are slightly different to some extent, but their combination into “SST-dependent” and “SST-independent” terms and the shifting of leader driver from SST-independent to SST-dependent are the same.

2. The authors proposed a multiple linear regression to distinguish the relative contributions of direct and indirect effects of heatwaves on NPP. I am uncertain whether the "direct effect" refers to thermal stress (i.e., anomalously high temperatures inhibit phytoplankton growth and cause mortality). If so, it is important to note that the relationship between phytoplankton growth rate and temperature is not linear, making it unsuitable for linear regression analysis. Additionally, the temporal evolution of phytoplankton and temperature does not follow the same pattern in many regions, with temperatures peaking in summer and reaching their lowest in winter, while phytoplankton peak in spring and are lowest in autumn and winter (Fernández-Barba et al., 2024; Li et al., 2024). The attribution of thermal stress to the NPP response to MHWs globally (Fig. 3) makes it challenging to reconcile the two contradictory results: NPP decreases in low latitudes and increases in higher latitudes (Fig. 1). As pointed out in the Introduction, previous studies have reported nutrient supply and light availability as the key factors influencing phytoplankton productivity. I recommend the authors perform an attribution analysis considering light and nutrients as the main drivers (e.g., examining the influences of the anomalies in the input variables on the estimated NPP (Le Grix et al., 2022; Silsbe et al., 2025), and explore in more detail how heatwaves affect light, mixing, nutrient supply, growth rates, and thus primary productivity.

Thank you for this important point. We acknowledge that ocean productivity during MHWs is governed by many interacting factors (changes in light availability, nutrient supply, mixing, etc.), so fully disentangling their roles requires detailed process models and is inherently complex (Le Grix et al., 2022). Our goal, however, was to leverage the observational record to identify broad empirical patterns in the response of NPP to warming. In our regression framework, the observed NPP anomaly is statistically partitioned into an “SST-dependent” component (the direct thermal effect plus coincident light changes and CHL change) and an “SST-independent” residual component (encompassing nonlinear responses). This decomposition exploits the empirical covariance structure in the data to highlight where temperature changes dominate productivity anomalies, without relying on a full ecosystem model. Crucially, the results are robust across independent NPP datasets and align qualitatively with process-based expectations. For example, Le Grix et al. (2022) show that tropical heatwaves tend to enhance nutrient limitation of phytoplankton, whereas high-latitude heatwaves more often lead to light limitation.

In the revised manuscript we have clarified that our regression analysis provides an empirical separation of SST-linked vs. SST-independent variability, and we explicitly acknowledge that a full mechanistic attribution (e.g. via large-ensemble model experiments) is beyond the scope of the present study.

3. The analysis of the anomalies in mixed layer depth, nutrients, and CHL during heatwaves in LMEs is informative. The authors attributed the reduction in primary productivity in low latitude regions to thermal stress, and the increase in higher latitude productivity to higher baseline nutrient levels. However, this explanation is unconvincing. Although these regions have higher nutrient levels and deeper mixed layers, during MHWs, nutrients decreased, and the mixed layer shoaled. These changes obviously cannot lead to an increase in primary productivity. As pointed out earlier in the Introduction, phytoplankton in higher latitudes are mainly influenced by light conditions. I suggest the authors include an analysis of PAR anomalies to determine whether it has a significant impact on the increase in productivity. From my understanding of the results, in the low latitudes that are nutrient-limited, the reduction in nutrient supply may possibly be responsible for the decline in productivity. As for the more pronounced reduction in nutrient levels compared to the higher latitudes, it is likely due to the higher baseline nutrient levels in the higher latitudes. Given that previous studies have extensively analyzed the effects of MHWs on chlorophyll, and the

primary productivity results in this study are closely related to those findings (Fernández-Barba et al., 2024; Noh et al., 2022; Zhan et al., 2024), I suggest the authors refer to their mechanistic analyses for further insight. I suggest the authors include an analysis of PAR anomalies to determine whether it has a significant impact on the increase in productivity.

Thanks for your comment. Indeed, our results are consistent with your interpretation: in the nutrient-limited low latitudes, the reduction in nutrient supply during MHWs contributes to the decline in productivity, whereas in high latitudes, the smaller nutrient reduction and higher baseline nutrient levels help sustain productivity. Details are provided in the updated Fig.4.

Following your suggestion, we have added an analysis of PAR anomalies in Fig. R2. However, the direct PARA contribution to NPPA is relatively small, and most of the variability appears in the nonlinear residual of the light-related processes (Fig. R2), reflecting the combined influences of nutrients, light availability, and phytoplankton physiological adjustments. In addition, we have included the spatial mean patterns of nutrients and MLD (Fig. S9), which further demonstrate the key distinction between low- and high-latitudes: high-latitude regions maintain much higher baseline nutrient levels. In our revised manuscript, we individually analyzed the MHW-enhanced NPPA regions (mostly high latitudes) and MHW-inhibited NPPA regions (mostly low latitudes) as show the MHW-inhibited regions with lower baseline nutrients and more sensitive to thermal forcing.

Here is the NPPA–PARA regression model for Fig R2, but NPP factors of SSTA and CHLA were regressed onto the PARA-related components ($SSTA_{par}$ and $CHLA_{par}$) and PARA-unrelated components ($SSTA_{res}$ and $CHLA_{res}$) as shown in Eq. (R1):

$$SSTA = \underbrace{\alpha PARA}_{SSTA_{par}} + SSTA_{res}, \quad CHLA = \underbrace{\beta PARA}_{CHLA_{par}} + CHLA_{res}, \quad (R1)$$

$$NPPA = \underbrace{aPARA + bSSTA_{par} + cCHLA_{par}}_{PAR\text{-dependent NPPA}} + \underbrace{dSSTA_{res} + eCHLA_{res} + NPPA_{res}}_{PAR\text{-independent NPPA}}. \quad (R2)$$

It is obviously that most variability appears in the nonlinear residual, rather than the direct influence of light (Fig R2e-f). We agree that NPP change are closely related to MLD, nutrient, and

light, which we have updated information in Fig. 4 and Fig. S9, as well as discussion it in Section ‘Reasons driving global ocean responses to MHWs’.

Fig R2 | Evaluation of the NPPA Mathematical Model: Linear regression of NPPA and PARA. (a) The R-squared value of the regression model. Panel (b–f) are regression coefficients of the individual parameters (a , b , c , d , and e) in Eq. (6). The contributions of each term during MHW periods are represented in Panels g-l.

4. The study presented a rich spatial variability of NPP anomalies during MHWs across the global ocean, while the analysis and discussion focused primarily on the LMEs. As shown in the Figures, large areas in the Southern Ocean, eastern tropical Pacific, tropical Atlantic, North Atlantic, and polar regions exhibit strong productivity responses to heatwaves, making them of great interest to a broad audience. If the authors prefer to focus on the LMEs, I recommend removing the results in the open oceans to present a clearer picture of LMEs and revising the title accordingly. However, if the authors wish to retain the open ocean results, I suggest modifying the manuscript to include a more detailed discussion of the characteristics and mechanisms of NPP response to MHWs across different major ocean regions.

Thank you for raising this important point. We agree that the strong NPP responses observed in the Southern Ocean, eastern tropical Pacific, tropical Atlantic, North Atlantic, and polar regions warrant discussion beyond the LMEs. For this reason, we chose to retain the open-ocean results in the revised manuscript and expanded the analysis accordingly.

Specifically, we now divide the global ocean into major dynamical–biogeochemical regions following the definitions in Fig. S8, and we provide new results for the Southern Ocean, subtropical gyre interiors, western boundary currents and their extensions, eastern boundary upwelling systems, and the tropical Pacific (revised Fig. 2j and Supplementary Table 2). Additional text has been added in Lines 88-92 and 140-145 to describe the distinct characteristics and mechanisms of NPP responses to MHWs across these regions.

Furthermore, Fig. 4 (Fig. R3) has been revised to global ocean rather than only LMEs, and we classify the global ocean into regions where NPP anomalies show positive or negative correlations with SSTA at the 90 percent confidence level, avoiding any ambiguity associated with the previous “MHW-enhanced” and “MHW-inhibited” terminology. This new framework enables a clearer interpretation of how MHWs influence productivity in different oceanic regimes and provides the broader perspective requested by the reviewer.

Fig R3 | Statistical contrasts between MHW-enhanced and MHW-inhibited regions. (a) Spatial distribution of the correlation between NPPA and SSTA, with green shading indicates positive correlations and brown indicates negative correlations. Regions where NPPA–SSTA correlations are significantly positive (confidence level >90%) are classified as MHW-enhanced regions, and significantly negative correlations define MHW-inhibited regions. LMEs are outlined and separated into MHW-enhanced LMEs (red boundaries) and MHW-inhibited LMEs (blue boundaries). In panels **b-o**, MHW-enhanced regions is labeled “Enhance” (red bars) and MHW-inhibited regions is labeled “Inhibit” (blue bars); black points denote group means and whiskers show interquartile ranges. (**b, c**) baseline mean state of SST and SST anomalies during MHWs; (**d, e**) baseline MLD and anomalies during MHWs; (**f, g**) baseline chlorophyll concentrations and anomalies during MHWs; (**h, i**) baseline PAR and anomalies during MHWs; (**j, k**) baseline nitrate (NO₃⁻) concentrations and anomalies during MHWs; (**l, m**) baseline phosphate (PO₄³⁻)

concentrations and anomalies during MHWs; (**n**, **o**) baseline silicate (Si) concentrations and anomalies during MHWs.

Minor Comments:

1. Lines 89-90: The NPPA during MHWs is also prominent in the Southern Ocean, the North Atlantic, and especially the polar regions.

Thanks. We added this clarification in Lines 92–93.

2. Lines 91-92: These statistical values are incredibly high and not supported by Figs. 1a and 1c, as NPP in most regions is less than 150, and the corresponding anomalies are below 25%. Please recheck these values.

Thanks. The sentence in question refers specifically to LMEs, which are characterized by much higher baseline NPP compared to the global average. Therefore, the reported values are correct within those regions.

3. Lines 100-101: The value cannot be that high as presented in Figs. 1c and 1f. Please recheck these values.

Thank for pointing this out. We have added Fig. S2 to quantify the difference between latitudes, and updated the version as “Specifically, MHW result in 4% to 21% extra NPP increase in the high-latitudes (45-80°S , 45-80°N) and 4% to 10% NPP reduction in the low-latitudes (45°S - 45°N) (Fig. S2a).” in Lines 88-90.

4. Line 101: Fig. 1c?

Revised. Thanks.

5. Line 109: "Although rising SST may enhance marine productivity³²", this statement conflicts with the results presented above and in Fig. 1c. Please revise for clarity.

Revised. Thanks.

6. Lines 113-114: This method cannot separate the relative contributions of thermal forcing and ecological processes, as the linear thermal regression does not fully represent the thermal response. SST follows a Gaussian distribution, whereas CHL follows a log distribution.

We agree that a linear SST–NPPA regression cannot fully represent the true thermal response, nor can it mechanistically separate thermal forcing from ecological processes. This is because many ecological processes (e.g., stratification-driven nutrient limitation, chlorophyll adjustments, trophic interactions) are themselves temperature-dependent and do not follow the same statistical distribution as SST (e.g., the log-normal behavior of CHL). Consequently, thermal and ecological pathways cannot be cleanly isolated using a linear model.

Our intention is not to identify each ecological process individually or to infer a mechanistic decomposition. Instead, as clarified in the revised text, our approach provides a statistical partitioning of NPPA variability into two components (Lines 113–119): “SST-dependent processes represent the linear covariation between SSTA and NPPA expressed through multiple pathways (hereafter referred to as the ‘thermal effect’), whereas SST-independent processes encompass all remaining variability, including nonlinear ecological responses to temperature, time-lagged effects, and any processes not directly covarying with SSTA (see ‘Statistical modeling of NPPA’ in Methods).”

7. Lines 125-130: This explanation does not make sense from a physical or ecological standpoint. MHWs are essentially more intense positive temperature anomalies, and the drivers of MHWs and SSTA+ are generally the same. Then, how could this shift in dominance occur? What is the underlying mechanism?

Thanks for this interesting question. Although both SSTA+ and MHWs represent positive temperature anomalies, their ecological and biogeochemical consequences differ in several key ways that lead to shifts in the relative contributions of the SST-dependent and SST-independent terms in Eq. (2).

First, MHWs effectively represent extreme temperature levels, not just stronger anomalies. Such extreme heat can induce nonlinear ecological responses—including pigment degradation, partial phytoplankton mortality, or physiological breakdown—that may not simply be proportional to

SSTA. These processes therefore appear in the SST-independent (residual) term rather than in the linear SST-dependent term. Please see the CHL response to MHWs and SSTA+ in Fig. R4.

Fig R4 | (a) and (c) represent Change of $CHLA_{sst}$ and $CHLA_{res}$ during $SSTA^+$ period, respectively. (b) and (d) are corresponding results but during MHWs.

Second, many MHWs are associated with anomalously shallow mixed layers, which suppress nutrient resupply from below. This mechanism reduces chlorophyll and modifies CHL_{res}, thereby changing the SST-independent contribution. Such stratification-driven effects are typically weak or absent during moderate SSTA+ events, which explains the shift in dominance between the two regimes.

Finally, the term “SST-dependent” may sound like it refers purely to direct thermal forcing, but as clarified in our revised text (Response to Comment 6), it represents the linear SSTA–NPPA covariation across pathways, whereas nonlinear, threshold-type, or time-lagged ecological responses fall into the SST-independent term. This definition is used consistently throughout the manuscript.

Considering that the sentence you refer to appears in the Results section, we have added the corresponding mechanistic explanation in the “Discussion” section, where interpretation of the underlying processes is more appropriate.

8. Line 137: “-30 to 100” should be “-30 to 50”?

Corrected. Thanks.

9. Line 142: Please add a reference of “(Fig. 2j)” after “respectively”.

Revised. Thanks.

10. Line 142: “During MHWs, this reverses dramatically: only 7 LMEs retain SST-independent dominance.” How does this reversal occur?

Revised in Lines 164-171. We have added a sentence after that: “Notably, the 7 LMEs that retain SST-independent dominance during MHWs are those where nonlinear ecological processes continue to exert a primary influence on NPPA, indicating that the response of these ecosystems remains strongly shaped by local, non-thermal dynamics.” The reasoning is related to the MLD-Nutrient relationship discussed in section “Reasons driving global ocean NPP response to MHWs”.

11. Lines 155-156: “We classified 54 global LMEs based on the correlation between SSTA and NPPA.” It would be better to say: “We classified the 54 global LMEs into two dominant categories based on the mean correlation between SSTA and NPPA for each region”?

Revised. Thanks.

12. Lines 176-177: The explanation for the mechanism is not convincing, as both MLD and nutrient anomalies are negative, and CHL does not show significant positive anomalies. Moreover, SST increase (thermal stress) exceeds the inhibition case.

Thank you for pointing this out. We have updated Fig. 4 to more broadly examine the global ocean rather than just LMEs, which show CHL increased in MHW-enhanced regions during

MHWs while reduced in MHW-inhibited regions. Please see the revised manuscript Lines 168-212.

13. Line 178: But why is there a positive Fe anomaly, contrasting with nitrate anomalies?

A positive Fe anomaly during MHWs, contrasting with the negative nitrate anomaly, arises from the distinct baseline distributions of the two nutrients. As shown in Fig. R5g, elevated Fe concentrations occur primarily in iron-rich coastal upwelling zones and marginal seas (e.g., the Peru–Chile upwelling, the Somali upwelling system, and the Indonesian seas), where Fe is strongly supplied by sediment resuspension and shelf processes. In these regions, MHW-induced shoaling of the mixed layer can entrain Fe-rich waters from the upper shelf and increase surface Fe concentrations.

In contrast, nitrate-rich waters are mainly located in high-latitude regions (Fig. R5d), where MLD shoaling during MHWs reduces vertical nutrient supply from deeper layers, producing widespread negative nitrate anomalies (Fig. R5f). Thus, the opposing signs of Fe and nitrate anomalies reflect their contrasting baseline distributions and the different ways in which MLD shoaling influences their vertical supply pathways.

Fig R5 | Spatial pattern of MLD and Nutrients during $SSTA^+$ and MHWs. (a) spatial pattern of mean MLD during 1988–2018. (b) and (c) are MLD changes during $SSTA^+$ periods and MHWs, respectively. (d-f) are mean nitrate concentrations (NO_3^-) and their corresponding changes during $SSTA^+$ and MHW periods. (g-i) are mean Fe concentrations and their corresponding changes during $SSTA^+$ and MHW periods.

14. Lines 181-183: It is important to clarify how shoaling mixing and reduced nutrient supply could lead to an increase in NPP.

Thank you. In the revision, we have clarified the mechanisms in the section “Reasons driving global ocean NPPA responses to MHWs”. Briefly, an increase in NPP during MHWs can occur despite shoaling MLD and reduced nutrient supply because these responses depend strongly on the baseline ecological state of each region. In nutrient-rich, high-productivity systems, the remaining nutrient concentrations after MLD shoaling often remain above limiting levels, allowing

warmer and more stratified conditions to enhance light utilization and support biomass accumulation. In addition, MHWs can promote shifts toward smaller phytoplankton groups with higher nutrient-use efficiency, enabling productivity to be sustained or increased even when nutrient inputs decline^{6,7}. These processes together explain how NPP may increase in certain regions under MHW conditions.

References

1. Thomas, M. K., Kremer, C. T., Klausmeier, C. A. & Litchman, E. A Global Pattern of Thermal Adaptation in Marine Phytoplankton. *Science* **338**, 1085–1088 (2012).
2. Eppley, R. Temperature and phytoplankton growth in the sea. *Fishery Bulletin* **70**, 1063–1085 (1972).
3. Sand, M. *et al.* Response of Arctic temperature to changes in emissions of short-lived climate forcers. *Nature Clim Change* **6**, 286–289 (2016).
4. Boyce, D. G., Lewis, M. R. & Worm, B. Global phytoplankton decline over the past century. *Nature* **466**, 591–596 (2010).
5. Behrenfeld, M. J. *et al.* Climate-driven trends in contemporary ocean productivity. *NATURE* **444**, 752–755 (2006).
6. Zhan, W. *et al.* Reduced and smaller phytoplankton during marine heatwaves in eastern boundary upwelling systems. *Commun Earth Environ* **5**, 629 (2024).
7. Silsbe, G. M., Fox, J., Westberry, T. K. & Halsey, K. H. Global declines in net primary production in the ocean color era. *Nat Commun* **16**, 5821 (2025).

Reply to Reviewer 1

We are very grateful to you for your time in carefully reading our manuscript and providing helpful comments that make our manuscript better. We have carefully considered each of your comments (in blue) and revised the manuscript accordingly. Please find our responses (in black) to your comments below.

Reviewer #1 (Remarks to the Author):

This revised version of the manuscript "Marine Heatwaves Shift Ocean Net Primary Productivity from the Tropics to the Poles" addresses all the concerns I raised in my first review. In particular, I appreciate the clarified distinction between SST-dependent and SST-independent processes, as well as the inclusion of two additional NPP algorithms.

Note that in the "Data Availability" section, l. 320, you should either remove "(Eppley-VGM)" or replace with "(Eppley-VGM, CbPM, CAFE)".

Revised in Line 321. Thanks.

You have shown that thermal stress increasingly governs productivity patterns under extreme warming. Your findings are key to informing adaptive fisheries management and climate-resilient policy in a rapidly warming ocean.

I therefore recommend the publication of this manuscript.

Thanks.

Reply to Reviewer 2

Reviewer #2 (Remarks to the Author):

The authors have made extensive revisions of the original manuscript in response to my previous comments, and I find the manuscript much improved. However, I have one additional comment that the authors should address before the paper can be accepted. I appreciate the two additional analyses the authors added using the absorption-based CAFÉ NPP and the carbon-based CBPM NPP products to validate their decomposition of "SST-dependent" and "SST-independent" NPPA during MHWs. However, the implication of this decomposition (Equations 1-2) to the CAFÉ and CBPM NPP is somewhat obscure. While I understand that NPP in the real ocean is subject to estimation algorithms, this regression is not physically convincing, as SST is not an input factor in the CAFÉ and CBPM NPP models. To avoid potential controversy, I suggest the authors replace this validation with a direct comparison of the consistency of NPP estimated from VGPM, CAFÉ, and CBPM products during MHWs.

We agree that SST is not an explicit input variable in the CAFÉ and CBPM NPP models, and therefore the SST-dependent term in Eqs. (1–2) should not be interpreted as representing a mechanistic temperature control within these models. To avoid potential misunderstanding and address this concern directly, we have revised the Supplementary analysis and no longer apply the SST-dependent/SST-independent decomposition to the CAFÉ or CBPM NPP products.

Instead, following the reviewer's suggestion, we now present a direct comparison of NPPA derived from the VGPM, CAFÉ, and CBPM products during MHWs (Fig. S3; Line 94-97; Line 264-265), focusing on the consistency of their responses across products that rely on fundamentally different assumptions and inputs. This revision avoids potential controversy associated with regression-based attribution in models for which SST is not an explicit driver.

In addition, we have revised the wording throughout the manuscript to clarify that the term "SST-dependent" refers to a correlation-based grouping of MHW-related processes, rather than implying a single-factor or mechanistic temperature control (Line 212-223).

We believe this revision strengthens the robustness and clarity of our conclusions while maintaining physical consistency across NPP products.